# Health-Related Quality of Life and Its Determinants among Criminal Police Officers

**DOI:** 10.3390/ijerph16081398

**Published:** 2019-04-18

**Authors:** Xinrui Wu, Qian Liu, Qi Li, Zhengwen Tian, Hongzhuan Tan

**Affiliations:** Department of Epidemiology and Health Statistics, Xiangya School of Public Health, Central South University, Changsha 410078, China; xwu13@tulane.edu (X.W.); sean-6868@163.com (Q.L.); jdcynie@163.com (Q.L.); laura1009@163.com (Z.T.)

**Keywords:** HRQoL, EQ-5D-3L, criminal police officers, determinants of HRQoL

## Abstract

Criminal police officers are viewed as having a very tiring and stressful job, one that is closely correlated with work disability and other factors that might impair quality of life. Few studies have addressed the issue of health-related quality of life (HRQoL) in this population. Thus, this study aimed to assess the HRQoL of criminal police officers compared with the general adult population and identify determinants associated with HRQoL. Based on a cross-sectional study of 281 criminal police officers in China, we used the EuroQol five-dimension three-level (EQ-5D-3L) scale, the Self-Rating Anxiety Scale (SAS), and the Self-Rating Depression Scale (SDS) to collect data. Tobit regression models and logistic regression models were used to investigate factors associated with HRQoL. The average EQ-5D-3L index score and EQ-5D visual analogue set (EQ-5D VAS) score were 0.919 and 77.22, respectively (total comparable population 0.958 and 80.12, respectively). Anxiety/depression and pain/discomfort were the most frequently-reported problems. Lower HRQoL was associated with age, drinking alcohol, physical activity, injury on duty, and symptoms of anxiety or depression. These findings indicated that criminal police officers have poorer quality of life than the general adult population and that risk-oriented interventions should be implemented to improve the HRQoL of criminal police officers.

## 1. Introduction

As a workgroup with a special profession, police officers have great responsibility for maintaining the stability of law and order in society. Notably, the work of criminal police officers involves repetition and long-term heavy workloads, frequent work shifts, intense work pressure, and threats of violence, all of which make criminal police officers more vulnerable to adverse psychological and physiological outcomes [1,2]. Compared with the normal population, previous studies among police officers have been scarce and have demonstrated that they disproportionately suffer from numerous health problems, including dyslipidemia, diabetes, metabolic disorders, anxiety, depression, suicide, and sleep disorders [3,4,5,6,7], which significantly decreases their health condition and work efficiency.

Health-related quality of life (HRQoL) is a well-known health condition representing individuals’ beliefs, experiences, and expectations in their daily life [8]. HRQoL includes not only physical health, but also mental and social well-being [9]. Previous research has demonstrated that HRQoL is associated with sociodemographic factors [10], lifestyle behaviors [11], social support [12], chronic disease history [13], mental health [14], and so on, in different study populations. HRQoL is an essential component of evidence-based public health policy that can be used to evaluate health care quality, analyze cost utility effectiveness, and assess the influence of diseases, disabilities, and disorders over time [15,16].

Criminal police officers have been reported to have a very tiring and stressful job, and the actual level of their HRQoL and its influencing factors remain to be explored. However, studies focusing on the HRQoL of criminal police officers are rare. Furthermore, to date, very few studies have explored the determinants contributing to HRQoL among criminal police officers; only Da Silva et al. examined the relationship between regular physical activity and the HRQoL of criminal police officers in Brazil by using the 36-Item Short Form Health Survey (SF-36) questionnaire [17].

During the past few decades, hundreds of questionnaires assessing HRQoL have been developed, including specific instruments (e.g., Diabetes Quality of Life (DQOL), Functional Assessment of Cancer Therapy (FACT)) and generic instruments (e.g., SF-36, Nottingham Health Profile (NHP)) [18,19]. Among the instruments that can be used with the general public and people with various health statuses, the EuroQol five-dimension three-level (EQ-5D-3L) scale is perhaps the most commonly used by researchers and governmental agencies [20,21,22,23,24]. Thus, comparison of HRQoL between a special population group with the general public and evaluating its health condition are possible [20,25,26]. The objective of our study was to examine the health status of criminal police officers in China by using the EQ-5D-3L as a measure of HRQoL and to explore the determinants of HRQoL in criminal police officers compared with the general adult population.

## 2. Methods and Materials

### 2.1. Study Population and Data Collection

Changsha is the capital of and the largest city in Hunan province in south China, and it has a population of over 7 million. Since criminal police represent a special occupational population, we selected all the criminal police officers working in the police divisions at and above the county level of Changsha city from March 2016–January 2017 for our study (*N* = 300). The subjects were interviewed to collect HRQoL, anxiety and depression symptoms, sociodemographic information, health-related behaviors, social and work conditions, and chronic disease history. All participants were interviewed face to face in a meeting room in their police station by trained interviewers who were postgraduate students and staff from the Xiangya School of Public Health, Central South University. Completed questionnaires were carefully checked by quality supervisors, and the data were double input after the interviews each day.

### 2.2. Measures

#### 2.2.1. The EQ-5D-3L

HRQoL was assessed using the EQ-5D-3L, an extensively-used, generic instrument for interviewees aged over 18 years [27]. The EQ-5D-3L has good sensitivity and acceptability and can reflect comprehensive health-related information of respondents on multiple dimensions [28]. This scale has shown good reliability and validity in its Chinese version [29,30]. It consists of two parts: the EQ-5D-3L questionnaire and the EQ-5D visual analogue set (EQ-5D VAS).

The EQ-5D-3L questionnaire contains five dimensions: mobility, self-care, usual activities, pain/discomfort, and anxiety/depression, and each dimension has three options: no problems, moderate problems, and extreme problems [27]. This questionnaire can be used to define up to 243 health statuses. The EQ-5D-3L index score, which ranges from −0.149–1.000, can be used to define health conditions according to the Chinese population-based time-trade model (TTO) [31]. Closer to 1.000 means a better HRQoL.

The EQ-5D VAS was presented on a horizontal line like a thermometer. The respondents were asked to point out which score represented their own health status based on their subjective health perception. The scores ranged from 0–100, and 0 was labeled “the worst imaginable condition of health”, while 100 was labeled “the best imaginable condition of health”. The higher the score was, the better the health condition [32].

#### 2.2.2. The SAS and the SDS

The Self-Rating Anxiety Scale (SAS) [33] and the Self-Rating Depression Scale (SDS) [34] were used to measure the level of anxiety and depression. They are two norm-referenced scales to reflect the subjective feelings of subjects with anxiety tendencies or depression severities in the past seven days, respectively. The reliability and validity of the Chinese version of these two scales have been confirmed in a previous study [35]. Both of them contained 20 items, and each item was classified as none or a little of the time, sometimes, often, and most of the time, which had an assigned score from 1–4, respectively. The testing score was calculated by summing the scores for the 20 items and was standardized by multiplying the sum by 1.25 (full standard scores ranged from 25–100). Higher scores on the SAS or the SDS indicated a higher level of mental disorder. According to the Chinese norm, a total standard score of 50 or 53 was set as a cut-off point of anxiety or depression, respectively [36].

#### 2.2.3. Variables Definition

Sociodemographic factors included sex, age (<30, 30–39, ≥40), ethnicity (Han, others), education (bachelor’s or below, master’s or above), and monthly income per person (<300, 300–449, 450–599, ≥600 USD). Health-related behavior factors included smoke, alcohol, sleep, and exercise. Smoke, alcohol, and sleep were coded as “yes or no” by asking the questions “Have you ever smoked?”, “Have you drunk alcohol during the last month?”, and “Have you stayed up late during the last month?”, respectively. Exercise, which represents physical activities per week, was recorded as little, sometimes (1–2), and regular (≥3). Social relationship and work condition factors referred to family relationships, peer relationships, and injuries on duty, which were identified by bad or good and yes or no. The chronic disease history was defined as “yes” when the participants were reported to have at least one disease comprising of hypertension, hyperlipidemia, diabetes mellitus, liver disease, renal disease, or gastrointestinal disease. 

### 2.3. Ethical Approval

This study was submitted for consideration and approved by the Medical Ethics Committee of Xiangya School of Public Health, Central South University (XYGW-2016-34). The investigation was conducted according to the principles of the Declaration of Helsinki and its amendments. All participants voluntarily participated in the research and filled out an informed consent form.

### 2.4. Data Analysis

First, we did some descriptive analysis. Due to the type and distribution of the variables, the mean ± standard deviation, median (interquartile range), and frequency (percentage) were used for the EQ-5D-3L index, the EQ-5D VAS, and people reporting any problems in any of the five dimensions of the EQ-5D-3L, respectively. Due to the limited sample size, the number of respondents reporting extreme problems was relatively small, so we merged the groups that reported “moderate problems” and “serious problems” on the EQ-5D-3L dimensions into an integrated group “have problems”. Then, we conducted univariate analysis. Mann–Whitney U tests (Kruskal–Wallis tests if more than two groups) were used for the analysis of the EQ-5D-3L index. Chi-square tests were conducted to analyze the categorical variables. Next, we conducted multivariate analyses to detect the effect size of the HRQoL-related factors after controlling for potential confounding factors. Variables with *p* ≤ 0.05 in the univariate analysis were incorporated into the multivariate analysis. Because the EQ-5D-3L index was a censored variable that showed a significant “ceiling effect” (a large proportion of respondents had the index score of one), we used the Tobit regression model to analyze the factors of the EQ-5D-3L index as Austin et al. suggested [37]. Finally, considering that anxiety/depression and pain/discomfort were the most frequently-reported problems that might result in poor HRQoL, we used logistic regression models to analyze the factors related to these two dimensions.

All analyses were conducted in Stata/MP 13.1 and SPSS 22.0 software, and *p* ≤ 0.05 was considered statistically significant.

## 3. Results

### 3.1. Sample Characteristics

Criminal police officers were excluded from the study for the following reasons: (a) not on active duty (*N* = 3); (b) retired because of illness (*N* = 2); (c) did not provide information due to a long-term business trip (*N* = 11); and (d) did not complete all the questionnaires (*N* = 3). Thus, the final sample size was 281. The average age of the respondents was 34.9 years (SE: 7.46 years old). Other basic information about the participants is shown in Table 1.

### 3.2. The EQ-5D-3L

The mean EQ-5D-3L index of the respondents was 0.919 ± 0.102. Of the 281 participants, 151 received the maximum score, indicating “ceiling effects” in the EQ-5D-3L index. However, the average score of the EQ-5D VAS was 77.22 ± 13.25. Of the respondents, 31.3% reported having problems in the anxiety/depression dimension and 26.7% reported having problems in the pain/discomfort dimension. The number of people who had moderate or serious problems in the mobility, self-care, and usual activities dimensions were 11, 1, and 5, respectively.

### 3.3. Determinants of HRQoL

#### 3.3.1. Univariate Analyses

Table 1 shows that the EQ-5D-3L index score varied with some sociodemographic and health-related factors of the participants. Those who were older, drank alcohol, did not exercise, had bad relationships with family or peers, had an injury history on duty, or had symptoms of anxiety or depression had lower EQ-5D-3L index scores than their counterparts. Considering that anxiety/depression and pain/discomfort were the most frequently-reported problems that might result in poor HRQoL, we performed analyses in these two dimensions. As shown in Table 2, respondents who were minority ethnicities, had a lower level of education, exercised little, had bad relationships with family or peers, reported an injury history on duty, or had a history of chronic disease were more likely to report problems in the anxiety/depression dimension. In addition, respondents who were older, drank alcohol, stayed up late, had bad family relationships, had occupational injury, had a history of chronic disease, or had symptoms of anxiety or depression were more likely to have problems in the pain/discomfort dimension.

#### 3.3.2. Multivariate Analyses

The results of the logistic regression on the anxiety/depression dimension showed that family relationship (OR = 0.480), peer relationship (OR = 0.210), and exercise (OR = 0.449) were positively related to good health condition and that injury on duty (OR = 2.481) was negatively related to good health condition for this dimension with other factors controlled (Table 3). The results of the logistic regression on pain/discomfort dimension showed that age (OR = 1.858), alcohol (OR = 1.973), and chronic disease history (OR = 4.340) were all risk factors for pain/discomfort with other factors controlled (Table 4).

The results of the Tobit regression model revealed that the determinants of the EQ-5D-3L index included age (β = −0.016), alcohol (β = −0.028), exercise (β = 0.026), injury history (β = −0.233), anxiety (β = −0.025), and depression (β = −0.037) with other factors controlled (Table 5).

## 4. Discussion

In the present study, we found that criminal police officers experienced lower HRQoL than the general adult population. Lower total HRQoL was associated with age, alcohol drinking, physical activity, injury on duty, and symptoms of anxiety or depression. The criminal police officers who had injury on duty were more likely to report problems in the dimension of anxiety/depression. Old age, drink alcohol, and chronic disease history were risk factors for the pain/discomfort dimension.

We found that the EQ-5D-3L index score for criminal police officers in Changsha city was 0.919, which was much lower than the general adult population, as in Hunan province (0.958) [38] and Heilongjiang province (0.959) [39] of China, and abroad, as in Singapore (0.95) [25] and Korea (0.94) [14]. The mean EQ-5D VAS score for criminal police officers (77.22) was also lower than the national average in China (80.12) [30] and Poland (79.6) [40], which indicates that more attention should be paid to exploring the reasons for this lower HRQoL and increasing quality of life in this special occupational group. However, the EQ-5D-3L index score for criminal police officers in Changsha city was higher than the norms reported in the USA (0.87) [23], Sweden (0.89) [41], and Demark (0.889) [42]. Sun et al. found that Asian people have more significant ceiling effects and higher grades on the EQ-5D-3L than other populations [30]; possibly due to the cultural difference that Asians are more likely to perceive a higher preference-based score than Whites, given the same health and disease conditions [43].

Our study identified some risk factors that were associated with HRQoL among criminal police officers. Older age was associated with poorer quality of life (β = −0.016), which corresponds to many studies in the general adult population [44,45,46]. Those who had injuries on duty had lower EQ-5D-3L index scores than those with no injury history (β = −0.233), which is similar to observations from Taiwan [47] and the USA [48]. Previous reports have found that injured policemen also have lower mental health scores, indicating that injury experience simultaneously affects physical and mental HRQoL on a large scale [49]. Thus, there is an urgent need for police departments to adopt institutional policies to ensure safe work and provide more health care after injury. Drinking alcohol was a risk factor for HRQoL, which is in line with the study conducted in Japanese workers [50]. This finding is partly because the association between alcohol use and problems in the pain/discomfort dimension (OR = 1.973), thus decreasing the EQ-5D-3L index score. Police officers who engaged in regular exercise had higher EQ-5D-3L index scores, which was consistent with Da Silva’s report [17]. There was a low level of regular physical activity among the policemen that could be due to physical and psychological overload during work [51]. 

Approximately 31.3% of the study participants reported problems in the anxiety/depression dimension, and 26.7% reported problems in pain/discomfort dimension, which was higher than that for the general adult population in China [52] and six European countries [53]. Our logistic regression analysis results showed that a good family relationship (OR = 0.480) and good peer relationships (OR = 0.210) had a protective effect on the incidence of problems reported in the anxiety/depression dimension. Working as a policemen involves a heavy workload, and consequently, police officers have little leisure time to spend with their family and friends. Furthermore, some adverse events such as violence, brutality, and death that police officers encounter in their daily work can lead to stress-related symptoms, thus compromising their relationships with peers and family [17]. These compromised relationships may result in less social support from family and peers, which can lead to more problems and influence police officers’ quality of life. We also found that chronic disease history (OR = 4.340) was a risk factor for criminal police officers’ problems in the pain/discomfort dimension, and this result is similar to those of previous studies conducted in the USA [54] and Thailand [55]. 

### 4.1. Strengths and Limitations

Our study has some strengths. First, this study revealed the current status of HRQoL in criminal police officers, which can provide an important population norm for the EQ-5D-3L in this specific occupational population. Second, this is the first study in China to investigate the influencing factors for lower HRQoL in criminal police officers. Some limitations must be addressed here. First, the study participants were recruited in only one city, which may limit the representativeness of the results. Second, due to the cross-sectional study design, we were unable to assess a chronological order between the risk factors and changes in HRQoL. Third, our investigation of health-related factors was retrospective and self-reported, which may cause recall bias.

### 4.2. Implications

These findings could be used to guide the allocation of public health resources to maximize cost-effectiveness in occupational disease prevention and health care interventions for criminal police officers. Firstly, the policymaker should pay close attention to physical examinations, ensure work safety, and provide more injury health care to prevent and control disease for criminal police officers. Secondly, measures such as decreasing workload, participating in outdoor activities, and providing health education should be adopted to allow police officers to engage in regular physical activity. Thirdly, the police department should pay more attention to police officers’ mental health, help them relieve themselves from adverse emotions, and get more social support from their peers and family with the purpose of improving their health, job performance, and quality of life.

For future research, a multicenter survey with a larger sample size and careful sampling design is needed to generate a more representative norm for the EQ-5D-3L in the criminal police officers’ population, and a longitudinal study is needed to confirm causal associations.

## 5. Conclusions

In summary, our study found that criminal police officers had lower HRQoL compared with the general adult population and that anxiety/depression was the most frequently-reported problem. Age, alcohol drinking, physical activity, injury on duty, and symptoms of anxiety/depression were important factors for decreased HRQoL. Our findings provide information for a better understanding of HRQoL in criminal police officers, and more effective risk-oriented interventions and health care policies might help prevent occupational diseases and promote health in this group.

## Figures and Tables

**Table 1 ijerph-16-01398-t001:** Characteristics of respondents in the EQ-5D-3L index and EQ-5D VAS.

		EQ-5D-3L Index				EQ-5D VAS
	*N* (%)	Median (IQR)/Mean ± SD	Ceiling Effect (%)	M-W/K-W	P(M-W/K-W)	Mean ± SD
**Total**	281	0.90 ± 0.10				77.22 ± 13.25
Sex						
Male	261 (92.9)	1.000 (0.131)	53.6			76.95 ± 13.49
Female	20 (7.1)	1.000 (0.130)	55.0	−0.424	0.671	80.70 ± 9.00
**Age**						
<30	81 (28.8)	1.000 (0.125)	69.1			80.32 ± 11.83
30–39	125 (44.5)	1.000 (0.131)	51.2			75.82 ± 13.45
≥40	75 (26.7)	0.875 (0.131)	41.3	17.371	0.001	76.15 ± 13.95
**Ethnicity**						
Han	269 (95.7)	1.000 (0.131)	52.4			77.07 ± 13.30
Others	12 (4.3)	1.000 (0.000)	83.3	−1.913	0.056	80.50 ± 11.96
**Education**						
Bachelor’s or below	259 (92.17)	1.000 (0.131)	51.4			77.29 ± 13.49
Master’s or above	22 (7.8)	1.000 (0.000)	81.8	5.539	0.063	76.41 ± 10.21
**Income**						
<300	38 (13.52)	0.875 (0.217)	42.1			75.75 ± 15.07
300–449	92 (32.7)	1.000 (0.130)	57.6			80.15 ± 11.00
450–599	128 (45.6)	1.000 (0.131)	53.1			76.30 ± 13.56
≥600	23 (8.2)	1.000 (0.131)	60.9	4.186	0.381	73.04 ± 15.10
**Smoke**						
No	129 (45.9)	1.000 (0.131)	58.1			77.15 ± 12.25
Yes	152 (54.1)	1.000 (0.131)	50.0	−1.409	0.159	77.28 ± 14.08
**Alcohol**						
No	203 (72.2)	1.000 (0.125)	57.6			78.01 ± 12.77
Yes	78 (27.8)	1.000 (0.217)	43.6	−3.052	0.002	75.17 ± 14.31
**Sleep**						
No	28 (10.0)	1.000 (0.131)	67.9			81.30 ± 8.33
Yes	253 (90.0)	1.000 (0.125)	52.2	−1.645	0.100	76.77 ± 13.62
**Exercise**						
Little	52 (18.5)	0.875 (0.217)	36.5			73.81 ± 11.42
Sometimes	148 (52.7)	1.000 (0.131)	53.4			77.63 ± 13.91
Regular	81 (28.8)	1.000 (0.125)	65.4	10.912	0.004	78.65 ± 12.88
**Family relationships**						
Bad	87 (31.0)	0.875 (0.217)	39.1			73.41 ± 11.92
Good	194 (69.0)	1.000 (0.131)	60.3	−3.535	0.000	78.93 ± 13.49
**Peer relationships**						
Bad	16 (5.7)	0.829 (0.293)	25.0			66.50 ± 13.45
Good	265 (94.3)	1.000 (0.131)	55.5	−3.035	0.002	77.87 ± 12.98
**Injury on duty**						
No	173 (61.6)	1.000 (0.125)	63.0			78.74 ± 12.76
Yes	108 (38.4)	0.875 (0.131)	38.9	−3.729	0.000	74.78 ± 13.71
**Chronic disease history**						
No	66 (23.5)	1.000 (0.125)	74.2			81.96 ± 13.73
Yes	215 (76.5)	0.875 (0.131)	47.4	−3.707	0.000	75.77 ± 12.78
**SAS**						
No	241 (85.8)	1.000 (0.131)	57.7			78.66 ± 12.63
Yes	40 (14.2)	0.869 (0.217)	30.0	−4.040	0.000	68.55 ± 13.73
**SDS**						
No	221 (78.6)	1.000 (0.131)	61.5			80.29 ± 11.91
Yes	60 (21.4)	0.875 (0.186)	25.0	−4.654	0.000	65.90 ± 11.78

Notes: The ceiling effect indicates the proportion of respondents with the best possible theoretical scores. The unit of income is USD (<2000, 2000–2999, 3000–3999, ≥4000 in RMB) SD: standard deviation, IQR: interquartile range, M-W/K-W: Mann–Whitney U tests or Kruskal–Wallis.

**Table 2 ijerph-16-01398-t002:** Percentage of reported problems on the pain/discomfort and anxiety/depression dimensions of the EQ-5D-3L.

	Pain/Discomfort			Anxiety/Depression		
	*N* (%)	χ^2^	*p*	*N* (%)	χ^2^	*p*
**Total**	75 (26.7)			88 (31.3)		
**Sex**						
Male	70 (26.8)			82 (31.4)		
Female	5 (25.0)	0.031	0.859	6 (30.0)	0.017	0.895
**Age**						
<30	10 (12.3)			20(24.7)		
30–39	37 (29.6)			44(35.2)		
≥40	28 (37.3)	13.401	0.001	24(32.0)	2.546	0.280
**Ethnicity**						
Han	73 (27.1)			88 (32.7)		
Others	2 (16.7)	0.644	0.422	0 (0.0)	5.716	0.017
**Education**						
Bachelor’s or below	72 (27.8)			86 (33.2)		
Master’s or above	3 (13.6)	2.079	0.149	2 (9.1)	5.482	0.019
**Income**						
<300	13 (34.2)			16 (42.1)		
300–449	21 (22.8)			26 (28.3)		
450–599	34 (26.6)			42 (32.8)		
≥600	7 (30.4)	1.966	0.579	4 (17.4)	4.662	0.198
**Smoke**						
No	31 (24.0)			39 (30.2)		
Yes	44 (28.9)	0.862	0.353	49 (32.2)	0.130	0.718
**Alcohol**						
No	44 (21.7)			60 (29.6)		
Yes	31 (39.7)	9.402	0.002	28 (35.9)	1.053	0.305
**Sleep**						
No	3 (10.7)			5 (17.9)		
Yes	72 (28.5)	4.057	0.044	83 (32.8)	2.619	0.106
**Exercise**						
Little	17 (32.7)			26 (50.0)		
Sometimes	42 (28.4)			48 (32.4)		
Regular	16 (19.8)	3.165	0.205	14 (17.3)	15.94	0.000
**Family relationship**						
Bad	30 (34.5)			40 (46.5)		
Good	45 (23.2)	3.911	0.048	48 (24.7)	12.592	0.001
**Peer relationship**						
Bad	7 (43.8)			12 (75.0)		
Good	68 (25.7)	2.523	0.112	76 (28.7)	15.052	0.000
**Injury on duty**						
No	35 (30.2)			42 (24.3)		
Yes	40 (37.0)	9.598	0.002	46 (42.6)	10.369	0.001
**Chronic disease history**						
No	5 (7.6)			13 (19.7)		
Yes	70 (32.6)	16.107	0.000	75 (34.9)	5.415	0.020
**SAS**						
No	55 (22.8)			-		
Yes	20 (50.0)	12.951	0.000	-	-	-
**SDS**						
No	50 (22.6)			-		
Yes	25 (41.7)	8.745	0.003	-	-	-

**Table 3 ijerph-16-01398-t003:** The results of the logistic regression on the anxiety/depression dimension.

Variables	B	S.E.	Wald χ^2^	*p*	OR (95%CI)
**Constant**	24.965	11087.2	0.000	0.998	
**Family relationships**	−0.734	0.298	6.058	0.014	0.480 (0.267,0.861)
**Peer relationships**	−1.559	0.639	5.955	0.015	0.210 (0.060,0.736)
**Exercise**	−0.801	0.214	13.994	0.000	0.449 (0.295,0.683)
**Injury on duty**	0.909	0.290	9.790	0.002	2.481 (1.404,4.383)

**Table 4 ijerph-16-01398-t004:** The results of the logistic regression on the pain/discomfort dimension.

Variables	*B*	*S.E.*	Wald χ^2^	*p*	OR (95%CI)
**Constant**	−4.846	1.151	17.716	0.000	
**Age**	0.62	0.191	10.479	0.001	1.858 (1.277,2.705)
**Alcohol**	0.679	0.317	4.598	0.032	1.973 (1.060,3.671)
**Chronic disease history**	1.468	0.509	8.330	0.004	4.340 (1.602,11.758)

**Table 5 ijerph-16-01398-t005:** The results of the Tobit regression on the EQ-5D-3L index.

Variables	S.E.	t	*p*	β (95%CI)
**Constant**	0.050	18.39	0.000	0.912 (0.814, 1.009)
**Age**	0.007	−2.46	0.014	−0.016 (−0.029, −0.003)
**Alcohol**	0.012	−2.36	0.019	−0.028 (−0.052, −0.005)
**Exercise**	0.008	3.36	0.001	0.026 (0.011, 0.042)
**Injury on duty**	0.011	−2.07	0.039	−0.233 (−0.046, −0.001)
**Anxiety**	0.011	−2.31	0.021	−0.025 (−0.047, −0.004)
**Depression**	0.011	−3.27	0.001	−0.037 (−0.060, −0.015)

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
