# Peer review of "Health-Related Quality of Life and Its Determinants among Criminal Police Officers"

_ijerph, 2019, doi:10.3390/ijerph16081398_

Round 1

Reviewer 1 Report

General

Authors explore in their manuscript ‘Health-related Quality of Life and Its Determinants among Criminal Police Officers’ Health-related Quality of Life by using the EQ-5D-3L scale and its determinants like anxiety and depression among criminal police officers in the PR China. HRQoL in healthy subgroups (employed persons) is understudied, as the phenomenon is most of the times studied in unhealthy subgroups (MS, PD, etc.), or that the group is divided along age, gender, education etc. However, some work should be done before this manuscript can be published.

Title

-

Abstract

Background

13           please delete <especially in China> - this is only relevant when a lot of studies have been done over the world and your idea is, that because of a certain country, culture or ethnicity, this might explain the difference you have in mind.

13           in the study aim the population you compare the HRQoL of police officers with, should be mentioned

Methods

14           before <We> put info on the type of study and the size of the sample.

Results

18           please indicate normal values behind <0.919 and 77.22, respectively> (total comparable population # and #, respectively)

It helps you to state (in your conclusion) that HRQoL among police officers is lower than that of the comparable population.

Conclusion

21           do you mean <the general population> (from 0-100) or do you mean <the comparable population> (I don't know exactly retirement age for policemen in PRC, but from 20-65)?

Key words

23           the last key word: shouldn't it be Determinants HRQoL

Introduction

30           please change the sentence <previous studies have demonstrated that police officers disproportionately> into <previous studies among police officers are scarce and have demonstrated that they disproportionately>. If the scarcity of the studies not would be stressed, then one might question why at all you do this.

51           <with the normal population>; do you mean <normal> or <age adjusted>?

54           <compared with the general population>; same question, do you want to compare a 20-65 group (police officers) with a group from 0-100 (general population)?

54           please delete or move to Implications sentence beginning with <These findings>

Methods

Sample

62           why are some districts separately mentioned after that you told us that you included all … of Changsha city? You mention <, including 4 police branches, namely, Tianxin District branch, Furong District branch, Yuhua District branch and Kaifu District branch,>

68-70     why are these two sentences here? <HRQoL was measured by the EQ-5D-3L Scale. The Self-Rating Anxiety Scale (SAS)[22] and the Self-Rating Depression Scale (SDS)[23] were used to measure the level of anxiety and depression.>

According to me they should be split (into a sentence on HRQoL and one on SAS & SDS) and belong to the next subheading.

Measures

92           Make a new line behind <condition[29].>; introduce both measures on anxiety and depression before <Both>

97+         you should say here how you measured (and where relevant how you treated; and if you used a cut-off point how you did) age, gender, ethnicity, education, income, smoking, consuming alcohol, sleeping, exercise, family relationship, peer relationship, injury, chronic disease history. In the Introduction you should already indicate why you (on the basis of references) might expect whether these variables could be determinants of HRQoL.

Statistical analyses

Please rewrite this section: First, we … . Then, we … . Next, we … . Finally, we … . The readership easier grabs what you did and in which order the Results will be shown.

Results

126-133 I would delete (as this information is already in Table 1) the following phrases:

<The average age of the respondents was 34.9 years (SE: 7.46 years old). Of the 281 subjects, 92.9% were men, and 95.7% were of Han ethnicity. Most of the respondents (91.1%) had a bachelor’s degree. The reported rates of smoking, drinking and staying up late were 54.1%, 27.8%, 90.0%, respectively. Only 28.8% of the respondents performed physical activities regularly. Of the respondents, 69.0% had a good relationship with their family, 94.3% had good relationships with their peers, and 38.4% had an injury history while on duty. For health condition, 76.5% of the participants had a history of chronic disease, 49.1% were overweight, 14.2% reported having symptoms of anxiety, and 21.4% reported having symptoms of depression>

128         Please, mind that you come with a brand-new variable, not present in Table 1: <staying up late>

<Income> in Table 1: please convert the valuta into something understandable for the readership, so USD, GBP, or EUR.

Same: income should be: <2000; 2000-2999; 3000-3999; ≥4000

Please explain (under Measures) how you divided yes/no of Smoking, Alcohol, Sleep (how did you measure this?), etc. etc. now I only can guess. 

137         please add to <3.2. EQ-5D-3L> also EQ-5D VAS

<Income> in Table 2: please have a look at what I said earlier regarding Table 1

167+       Table 3: the β should be a B?

168+       Table 4: the β should be a B?

Discussion

The Discussion section contains a discussion of the earlier presented findings.

Please harmonise the research question in the Abstract, in the Introduction section, and the first line of the Discussion section.

Please keep in mind the following structure for writing a Discussion:

para1                      start with repeating the research question + answer this without any comments or interpretation.

174         confirmed – you suggest that a similar study has been done before? Use We found 

175         general population. I mentioned this earlier; if you have to, because you don't have other data and you cannot recalculate those data, then at least you should mention this skewed comparison in the Limitations.

182         please remove the S from <Sun, S et al. found>

184         cultural differences and value sets – could you be a bit more specific?

Para2,3,4               start a new para, 1 topic per para, and start this para with one of your findings – which then defines the content of the para. Relate your finding to earlier published references.

196         Please delete <Lack of physical activity ranks fourth in the leading cause of global mortality[44], unfortunately,>

198         <resulting from> better would be <that could be due to>

204         <general population>; I already said something about this comparison

Strengths and limitations

Add subheading.

220         Did your study also have strengths? Please mention them, before going into the Limitations!!

Implications (split into: for practice, for future research)

Add subheading.

What does your study mean for practice or policy?

What do your findings mean for future research?

[remove last sentence of the previous para]

Conclusion

229         general population

Please rewrite this part, especially the first para, and add text and subheadings where they are absent

Tables, Figures

-

 References

Why use capitals in journal’s NAMES all over?

Why use capitals in ref 11, and 23?

Author Response

Response to Reviewer 1 Comments

Point 1: Line 13: (1) Please delete <especially in China> - this is only relevant when a lot of studies have been done over the world and your idea is, that because of a certain country, culture or ethnicity, this might explain the difference you have in mind.

(2) In the study aim the population you compare the HRQoL of police officers with, should be mentioned

Response 1: We are thankful for the reviewer’s comments. We have deleted the sentence “especially in China” (Revised line 13). We have mentioned “the general adult population” as the population which compared with police officers (Revised line 13-14).

Point 2: Line 14: Before <We> put info on the type of study and the size of the sample.

Response 2: We are thankful for the reviewer’s reminder. We have added some information about this study in the revised manuscript (Revised line 14-15).

Point 3: Line 18: Please indicate normal values behind <0.919 and 77.22, respectively> (total comparable population # and #, respectively). It helps you to state (in your conclusion) that HRQoL among police officers is lower than that of the comparable population.

Response 3: We appreciate the reviewer’s comment. We have added normal values about the comparable population in the revised manuscript (Revised line 19-20).

Point 4: Line 21: Do you mean <the general population> (from 0-100) or do you mean <the comparable population> (I don't know exactly retirement age for policemen in PRC, but from 20-65)?

Response 4: We appreciate the reviewer’s reminder. “The general population” here means the adults aged 18+ who can be the comparable population to the criminal police officers. The retirement age for policemen in PRC is 65 and as previous studies demonstrated age was considered as a influencing factor in our study. In order to make the description clearly, we have modified “the general population” to “the general adult population” in the whole revised manuscript (Revised line 23-24).

Point 5: Line 23: the last key word: shouldn't it be Determinants HRQoL

Response 5: We are thankful for the reviewer’s comment. We aimed to investigate the determinants associated with the lower HRQoL, so considering the reviewer’s suggestion, we have changed the last key word to “Determinants of HRQoL” (Revised line 26).

Point 6: Line 30: Please change the sentence <previous studies have demonstrated that police officers disproportionately> into <previous studies among police officers are scarce and have demonstrated that they disproportionately>. If the scarcity of the studies not would be stressed, then one might question why at all you do this.

Response 6: We are thankful for the reviewer’s comment and we have changed the sentence in the revised manuscript (Revised line 34).

Point 7: Line 51: <with the normal population>; do you mean <normal> or <age adjusted>?

Response 7: We appreciate the reviewer’s reminder. “The normal population” there means “normal” which is the general public and can be comparable to the specific study subjects. Following the reviewer’s suggestion, we have revised it to “the general public” in the revised manuscript (Revised line 58).

Point 8: Line 54: <compared with the general population>; same question, do you want to compare a 20-65 group (police officers) with a group from 0-100 (general population)?

Response 8: We are thankful for the reviewer’s comment and apologized for our incorrect expression. The EQ-5D-3L is a instrument designed to assess the HRQoL of interviewees aged over 18 years and many EQ-5D studies had been conducted in different areas which can be the comparable norm. We wanted to compare the criminal police officers with the adults aged 18+. So we have changed “the general population” to “the general adult population” in the whole revised manuscript (Revised line 61).

Point 9: Line 54: Please delete or move to Implications sentence beginning with <These findings>

Response 9: We are thankful for the reviewer’s comment and have moved the sentence to the “Implications” part (Revised line 328-330).

Point 10: Line 62: Why are some districts separately mentioned after that you told us that you included all … of Changsha city? You mention <, including 4 police branches, namely, Tianxin District branch, Furong District branch, Yuhua District branch and Kaifu District branch,>

Response 10: We appreciate the reviewer’s reminder. The reason why we mentioned all the districts’ name is to make the description of our study area more clearly. Also, we need to mention that this study was conducted in the police divisions at and above the county level of Changsha City, because the police department below the county level do not subdivide the policemen to criminal police officers. Following the reviewer’s suggestion, we changed these sentences to “in the police divisions at and above the county level of Changsha City” and deleted the sentences “including … ” in the revised manuscript (Revised line 70-72).

Point 11: Line 68-70: Why are these two sentences here? <HRQoL was measured by the EQ-5D-3L Scale. The Self-Rating Anxiety Scale (SAS)[22] and the Self-Rating Depression Scale (SDS)[23] were used to measure the level of anxiety and depression.>

According to me they should be split (into a sentence on HRQoL and one on SAS & SDS) and belong to the next subheading.

Response 11: We are thankful for the reviewer’s comment. These sentences should belong to the next subheading, so we have deleted them here (Revised line 78-80).

Point 12: Line 92: Make a new line behind <condition[29].>; introduce both measures on anxiety and depression before <Both>

Response 12: We appreciate the reviewer’s comment. We have rewritten this part and add some introduction to these two scales in the revised manuscript (Revised line 111-121).

Point 13: Line 97+: You should say here how you measured (and where relevant how you treated; and if you used a cut-off point how you did) age, gender, ethnicity, education, income, smoking, consuming alcohol, sleeping, exercise, family relationship, peer relationship, injury, chronic disease history. In the Introduction you should already indicate why you (on the basis of references) might expect whether these variables could be determinants of HRQoL.

Response 13: We are thankful for the reviewer’s comments. We have defined the variables in the “Measures” part (Revised line 123-133). We have added some influencing factors of HRQoL in the “Introduction” part (Revised line 40-42).

Point 14: Statistical analyses

Please rewrite this section: First, we … . Then, we … . Next, we … . Finally, we … . The readership easier grabs what you did and in which order the Results will be shown.

Response 14: We appreciate the reviewer’s reminder. Following the reviewers’ suggestion, we have rewritten this part in the revised manuscript (Revised line 141-157).

Point 15: Line 126-133: I would delete (as this information is already in Table 1) the following phrases: <The average age of the respondents was 34.9 years (SE: 7.46 years old). Of the 281 subjects, 92.9% were men, and 95.7% were of Han ethnicity. Most of the respondents (91.1%) had a bachelor’s degree. The reported rates of smoking, drinking and staying up late were 54.1%, 27.8%, 90.0%, respectively. Only 28.8% of the respondents performed physical activities regularly. Of the respondents, 69.0% had a good relationship with their family, 94.3% had good relationships with their peers, and 38.4% had an injury history while on duty. For health condition, 76.5% of the participants had a history of chronic disease, 49.1% were overweight, 14.2% reported having symptoms of anxiety, and 21.4% reported having symptoms of depression>

Response 15: We are thankful for the reviewer’s comment. But as the mean age was not listed in Table 1, we deleted this part except the first sentence “The average age of the respondents was 34.9 years (SE: 7.46 years old).” in the revised manuscript (Revised line 165-173).

Point 16: Line 128: (1) Please, mind that you come with a brand-new variable, not present in Table 1: <staying up late>.

(2) <Income> in Table 1: please convert the valuta into something understandable for the readership, so USD, GBP, or EUR. Same: income should be: <2000; 2000-2999; 3000-3999; ≥4000.

(3) Please explain (under Measures) how you divided yes/no of Smoking, Alcohol, Sleep (how did you measure this?), etc. etc. now I only can guess. 

Response 16: We appreciate the reviewer’s reminders. (1) Actually “staying up late” was represented by the variable “sleep” in Table 1 and we have explained this variable in the revised manuscript (Revised line 126-127). (2) Following the reviewer’s suggestion, we have transformed the number to USD in Table 1 and made note under the Table 1 (Revised line 176). (3) We apologized for the absence of the variables definition. We have explained those variables in the “Measures” part (Revised line 123-133).

Point 17: Line 137: (1)Please add to <3.2. EQ-5D-3L> also EQ-5D VAS

(2)<Income> in Table 2: please have a look at what I said earlier regarding Table 1

Response 17: We are thankful for the reviewer’s comments. (1) However, the EQ-5D-3L consists of two parts: the EQ-5D-3L questionnaire and the EQ-5D VAS, so the results of EQ-5D-3L index and EQ-5D VAS both belong to the EQ-5D-3L. After carefully consideration, we thought it might be better to remain the heading “The EQ-5D-3L”. (2) We have transformed the number to USD in Table 2 in the revised manuscript.

Point 18: Line 167+       Table 3: the β should be a B?

Line 168+       Table 4: the β should be a B?

Response 18: We appreciate the reviewer’s comment and apologize for our mistake. We have corrected them in Table 3 and 4 in the revised manuscript.

Point 19:

Discussion

The Discussion section contains a discussion of the earlier presented findings.

Please harmonise the research question in the Abstract, in the Introduction section, and the first line of the Discussion section.

Please keep in mind the following structure for writing a Discussion:

para1                      start with repeating the research question + answer this without any comments or interpretation.

174         confirmed – you suggest that a similar study has been done before? Use We found 

175         general population. I mentioned this earlier; if you have to, because you don't have other data and you cannot recalculate those data, then at least you should mention this skewed comparison in the Limitations.

182         please remove the S from <Sun, S et al. found>

184         cultural differences and value sets – could you be a bit more specific?

Para2,3,4               start a new para, 1 topic per para, and start this para with one of your findings – which then defines the content of the para. Relate your finding to earlier published references.

196         Please delete <Lack of physical activity ranks fourth in the leading cause of global mortality[44], unfortunately,>

198         <resulting from> better would be <that could be due to>

204         <general population>; I already said something about this comparison

Strengths and limitations

Add subheading.

220         Did your study also have strengths? Please mention them, before going into the Limitations!!

Implications (split into: for practice, for future research)

Add subheading.

What does your study mean for practice or policy?

What do your findings mean for future research?

[remove last sentence of the previous para]

Conclusion

229         general population

Please rewrite this part, especially the first para, and add text and subheadings where they are absent

Response 19: We are very thankful for the reviewer’s careful comments. According to the reviewer’s suggestion, we have rewritten this part and have modified all the problems you have mentioned in the revised manuscript (Revised line 267-338). And “the general population” here also means the adults aged 18+ who can be the comparable population to the criminal police officers. So we change it to “the general adult population” (Revised line 270-339).

Point 20:

 References

Why use capitals in journal’s NAMES all over?

Why use capitals in ref 11, and 23?

Response 20: We appreciate the reviewer’s reminders and apologize for our negligence. We have modified the format of references as required in the revised manuscript.

Reviewer 2 Report

This is an excellent article that I highly recommend for publication. Two elements would need to be developed: (1) It would be appreciated to have a little more comparisons at the international level. This would enrich the article all the more as the theme is rarely treated from this angle. (2) Recommendations would really be expected in the Discussion.

Author Response

Response to Reviewer 2 Comments

Point 1: This is an excellent article that I highly recommend for publication. Two elements would need to be developed: (1) It would be appreciated to have a little more comparisons at the international level. This would enrich the article all the more as the theme is rarely treated from this angle. (2) Recommendations would really be expected in the Discussion.

Response 1: We are thankful for the reviewer’s comments.

(1) We have rewritten the “Discussion” part and added more comparisons at the international level: compared the EQ-5D-3L index score with the Korea(Revised line 279) and Sweden(Revised line 283), compared the mean EQ-5D VAS score with Poland(Revised line 280), compared the risk factor “injury on duty” with the USA population(Revised line 292), compared “the reported questions in EQ-5D-3L dimensions” with six European countries(Revised line 304).

(2) We have added a subheading “Implications” in the “Discussion” part and made some suggestions there (Revised line 328-336).

Round 2

Reviewer 1 Report

Nice to read the manuscript now; i think it improved a lot and is more relevant for those scientists outside the PR China